# The evolution of ovary-biased gene expression in Hawaiian *Drosophila*

**Samuel H. Church**[1,2]*, **Catriona Munro**[3], **Casey W. Dunn**[2], **Cassandra G. Extavour**[1,4,5]

**1** Department of Organismic and Evolutionary Biology, Harvard University, Cambridge, Massachusetts, United States of America, **2** Current address: Department of Ecology and Evolutionary Biology, Yale University, New Haven, Connecticut, United States of America, **3** Collège de France, PSL Research University, CNRS, Inserm, Center for Interdisciplinary Research in Biology, Paris, France, **4** Department of Molecular and Cellular Biology, Harvard University, Cambridge, Massachusetts, United States of America, **5** Howard Hughes Medical Institute, Chevy Chase, Maryland, United States of America

* samuelhchurch@gmail.com

**Data Availability Statement:** All data are available at GitHub, under the repository 'shchurch/hawaiian_drosophilidae_expression_2021', commit '67d8e6f'. Both the full and strict (after removing outlier transcripts) correlation matrices

## Abstract

With detailed data on gene expression accessible from an increasingly broad array of species, we can test the extent to which our developmental genetic knowledge from model organisms predicts expression patterns and variation across species. But to know when differences in gene expression across species are significant, we first need to know how much evolutionary variation in gene expression we expect to observe. Here we provide an answer by analyzing RNAseq data across twelve species of Hawaiian Drosophilidae flies, focusing on gene expression differences between the ovary and other tissues. We show that over evolutionary time, there exists a cohort of ovary specific genes that is stable and that largely corresponds to described expression patterns from laboratory model *Drosophila* species. Our results also provide a demonstration of the prediction that, as phylogenetic distance increases, variation between species overwhelms variation between tissue types. Using ancestral state reconstruction of expression, we describe the distribution of evolutionary changes in tissue-biased expression, and use this to identify gains and losses of ovary-biased expression across these twelve species. We then use this distribution to calculate the evolutionary correlation in expression changes between genes, and demonstrate that genes with known interactions in *D. melanogaster* are significantly more correlated in their evolution than genes with no or unknown interactions. Finally, we use this correlation matrix to infer new networks of genes that share evolutionary trajectories, and we present these results as a dataset of new testable hypotheses about genetic roles and interactions in the function and evolution of the *Drosophila* ovary.

## Author summary

Gene expression data is commonly collected and compared across species. But to know which differences in gene expression are significant, we first need to know how much expression variation across species we expect to observe. Here we test this question by comparing RNA sequencing data across body parts and species of Hawaiian *Drosophila*

can be interactively visualized and queried at the accompanying data visualization for this paper (visualization at https://shchurch.shinyapps.io/hawaiian_fly_dataviz_2021 and repository at https://github.com/shchurch/hawaiian_fly_dataviz_2021). Raw RNA sequencing data are available at the Sequence Read Archive of the National Center for Biotechnology Information (NCBI), under BioProject PRJNA731506. Assembled transcriptomes and DNA barcode sequences are available at GitHub, under the repository 'http://github.com/shchurch/hawaiian_drosophilidae_phylogeny_2021', commit 'b12cbb10'. All code and results for this manuscript are available at GitHub, under the repository 'shchurch/hawaiian_drosophilidae_expression_2021', commit '67d8e6f'. The code to perform all agalma commands was performed in clean anaconda environment, installed following the instructions at 'https://bitbucket.org/caseywdunn/agalma'. All R commands were performed with a fresh install of R, and the session information including all package versions is available in the GitHub repository under the file 'r_session_info.txt'. The code to generate all plots as well as the text of this manuscript is available in several R scripts and Rmarkdown files at the same location.

**Funding:** This work was partially supported by National Science Foundation Graduate Research Fellowship Program (https://nsfgrfp.org/) DGE1745303 to SHC, National Institutes of Health award R01 HD073499-01 (NICHD) to CGE. (https://grants.nih.gov/grants/funding/r01.htm) The funders had no role in study design, data collection and analysis, decision to publish, or preparation of the manuscript.

**Competing interests:** The authors have declared that no competing interests exist.

flies, with a focus on genes expressed in the ovary. We identify a cohort of genes that have maintained ovary-specific expression over evolutionary time, and compare its composition to well-studied ovary genes in the laboratory model species *D. melanogaster*. We also show that, as the evolutionary distance between species being compared increases, variation between species overwhelms variation across tissues. We reconstruct the evolutionary history of genes that have changed expression patterns, and identify genes that have gained or lost expression in the ovary. We then use the dataset of evolutionary changes in expression to test for signatures of correlated expression evolution between genes. We show that genes that are predicted to interact show more correlated evolution than we expect by chance. Finally, we use the correlation data to generate new hypotheses about networks of genes that share evolutionary trajectories.

## 1 Introduction

Data on when and where genes are expressed are now fundamental to the study of development and disease [1]. With continually advancing RNA sequencing technologies, these data have been collected using RNA sequencing from a wide variety of cells, treatments and species [2, 3]. Statistical analysis of gene expression across these differentials generates insights into how gene expression is connected to phenotypic differences in morphology and behavior [4]. When comparing gene expression across species, most studies have relied on pairwise comparisons, often between one model laboratory species and one other species of interest [5]. One challenge with such pairwise comparisons is that they lack robust information about how much evolutionary variation in expression we expect to observe, making it difficult to evaluate the significance of any interspecific difference in variation [5, 6]. Phylogenetic comparisons of expression allow us instead to take into account the shared history between species [7, 8], and to describe significant changes in expression in relation to other phenotypic traits of interest [9]. In this study we perform a phylogenetic comparison of gene expression across the organs of twelve species of Hawaiian Drosophilidae flies with highly divergent ovary and egg morphologies. From our results we identify individual genes that have undergone significant evolutionary shifts in organ-specific expression, and describe global patterns in transcriptome variation across species that can serve as a benchmark for future interspecific comparisons of gene expression.

Phylogenetic comparisons of developmental traits are particularly valuable for building context around comparisons between well-studied model organisms and their non-model relatives [10]. Much more has been learned about the genetics and development of laboratory model species like *D. melanogaster* than may ever be possible for the vast majority of life [11]. But the usefulness of model species to understand general principles depends in part on the extent to which biology in these species reflects the biology of other taxa, rather than species-specific phenomena [12]. In the case of gene expression, there has been substantial debate about the degree to which patterns observed in model organisms may be representative across species [13–16]. Where several studies showed that the expression profiles of organs within a species are more different the profiles of homologous organs across species [17–20], other work has questioned this finding [13, 14]. More recently, Breschi and colleagues (2016) [21] demonstrated that, consistent with an evolutionary model of trait evolution, species-level variation in gene expression increases with the time since divergence from the most recent common ancestor. In addition, previous work by authors on this manuscript [8] showed that, while expression patterns across tissues tend to be consistent between species, lineage-specific

shifts in expression enrichment can be identified by applying phylogenetic comparative methods. With the exception of the work by Munro and colleagues (2021) [8], these studies have been, to our knowledge, performed almost exclusively in vertebrate species [17, 18, 20], and for the most part placental mammals [13, 14, 16], meaning that far less is known about organ and species-level expression differences when comparing across the tree of life.

The detailed atlases of expression data across organs [22] and developmental timepoints [23] is one of the strengths of model systems like *D. melanogaster*. These public resources make it possible to explore global patterns of expression to gain insight into potential gene regulation, interaction, and function [23–25]. As atlases such as these have become increasingly detailed and available from more taxa, a new goal has been to compare these expression profiles across species [7, 26, 27]. One objective of these cross-species comparisons is to shed light on potential regulatory associations between genes [7, 9]. This is especially advantageous for complex processes such as ovarian function for which we have a fragmented understanding of gene regulation despite genetic and transcriptome studies within single model organisms. Another objective of phylogenetic comparisons of expression atlases is to estimate the evolutionary distance between species at which we might expect a given gene to demonstrate a divergent pattern of expression [6]. If this distance is relatively small, then we predict atlases to contain large amounts of species-specific patterns. Alternatively, if as described above, variation across tissues outweighs variation across the species being compared, we predict atlases to contain large cohorts of tissue-specific genes that have been evolutionarily conserved. In this study we test for the existence of a core suite of ovary-specific genes across species of Hawaiian Drosophilidae and describe its size and composition in relation to the described atlas of expression in *D. melanogaster*.

The *Drosophila* ovary has several features [28] that make it a compelling organ in which to test hypotheses about expression evolution. Analyses of the FlyAtlas2 dataset [29] show that in *D. melanogaster*, more genes demonstrate highest expression enrichment in the ovary than any other adult female organ (Fig A in S1 Text). Additionally, all described signaling pathways are known to have a role in regulating ovarian development [30]. The ovary performs several critical functions, including maintaining the germ line and manufacturing specialized egg cells, yolk, and egg-shell materials [31]. Genetic screens [30, 32] and experimental manipulation in *D. melanogaster* have revealed functions of many genes involved in these processes, including yolk-protein genes required for oogenesis [33] and embryonic patterning genes with localized mRNA like *nanos* [34] and *swallow* [35]. Here we compare whole-ovary RNA profiles to assess the extent to which these genes and others demonstrate consistent patterns of ovary-enrichment over evolutionary timescales in a clade with highly divergent ovary and egg morphologies.

The Hawaiian Drosophilidae clade contains an estimated 1,000 extant species [36] that diverged from a common ancestor with *D. melanogaster* between 25 and 40 million years ago [37]. Extant species have been studied in particular for the variation in ovary and egg morphology [38, 39]. Species of Hawaiian Drosophilidae show the largest range within the family of egg size, shape, and the number of egg-producing units in the ovary, known as ovarioles [40–42]. Previous studies by our research group and others have shown that these traits are likely associated with evolutionary changes in the egg-laying substrate (e.g. rotting bark, flowers, leaves) [38, 40]. Furthermore, our previous work demonstrated that at least one developmental process, governing how the number of ovarioles is specified in the adult *D. melanogaster* ovary, is conserved in Hawaiian *Drosophila* [40]. The diversity of Hawaiian species and their relationship to model species make them a strong candidate model clade for evo-devo research [36, 43]. However, their relatively long generation times and species-specific breeding requirements make laboratory culture more challenging than classic *Drosophila* models [36]. In this

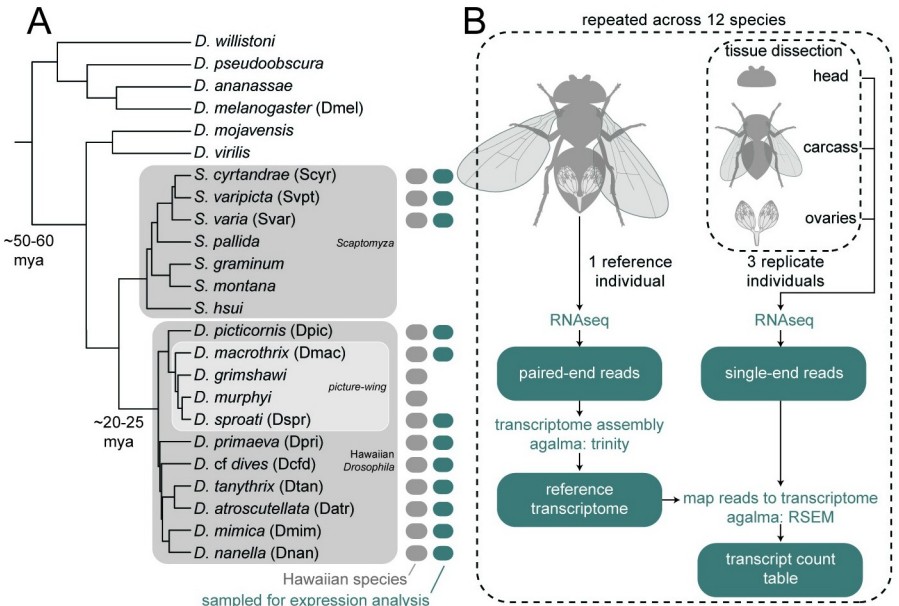

**Fig 1. Phylogeny of species and RNA sampling strategy.** A, Twelve species of Hawaiian Drosophilidae flies were collected in the wild and processed for RNA sequencing. The twelve reference transcriptomes assembled from these species were combined with twelve published genomes to generate the phylogeny shown here (originally published in Church and Extavour, 2021 [44]). Divergence times estimated by Russo and colleagues, 2013 [45]. Three clades within the group are highlighted: the genus *Scaptomyza*, nested within the paraphyletic genus *Drosophila*; the Hawaiian *Drosophila*, which, along with *Scaptomyza*, make up the Hawaiian Drosophilidae; and the well-known *picture-wing* clade. Adjacent to tip labels are four letter species codes used throughout the manuscript. B, The experimental design used to generate the data in this manuscript. When sufficient specimens were available per species, one whole individual was used as a reference and three whole individuals were dissected into three separate tissues: the head, ovaries, and all remaining material (carcass). Reference individuals were sequenced to generate paired-end RNA reads and dissected tissues were sequenced to generate single-end RNA reads. Tissue libraries were then mapped to the assembled reference to quantify transcript expression. Teal boxes indicate data files. Dashed-line boxes indicate a repeated step.

study we leverage technologies that can be deployed on wild-caught individuals to gather rich developmental data to compare across species.

We compared the expression profiles of twelve species of wild-caught Hawaiian Drosophilidae species across three body parts: the adult ovary, head, and the remaining carcass (Fig 1). We use these tissues to make two comparisons for calculating differential expression: one between the ovary and the carcass, and the other between the head and the carcass. These comparisons allow us to assess ovary-specific and head-specific gene expression over evolutionary time, both of which are relevant to fundamental questions in Hawaiian Drosophilidae biology. Throughout the text we use ovary-specific and head-specific to refer to transcripts that are significantly differentially upregulated in the ovary or head relative to the carcass. We use the term ovary-biased and head-biased to refer to transcripts that have a positive ratio of expression between the ovary or head and carcass, regardless of magnitude. The use of expression ratios is advantageous as potential effects of variation across individuals (e.g. in transcript counting efficiency) will be present in both the numerator and denominator, and therefore canceled out [7]. Given that the ovary and head are not equivalent body parts in terms of functional complexity (the ovary is primarily dedicated to producing oocytes, while the head contains the eyes, brain, and mouthparts, all dedicated to different tasks), we present analyses on these two body parts in parallel, and do not draw conclusions based on direct comparisons

between the head and ovary. In our description of results we prioritize the ovary-carcass comparison.

For each analyses, we first characterized the differentially expressed genes in the ovary and head of each species individually. By comparing these to each other, and to records of ovary-enriched and head-enriched genes from *D. melanogaster*, we identified core suites of tissue-specific genes shared across species. We applied linear modeling to this dataset to test the overall contribution of species- and tissue-level differences to expression variation across genes, and describe the circumstances under which one is likely to dominate over the other. Finally, we used a phylogenetic analysis of expression changes over evolutionary time to identify genes likely to have gained and lost tissue-enriched expression. This evolutionary screen of expression changes allowed us to identify networks of genes that demonstrate correlated changes in expression evolution. We provide these networks as a searchable dataset of novel, testable hypotheses for gene regulation with respect to ovarian function. The results of this study demonstrate both the power of Hawaiian *Drosophila* as a model clade for evo-devo, and the potential of using phylogenetic methods to identify evolutionary variation in gene expression underlying phenotypic differences.

## 2 Results

### 2.1 Differential gene expression reveals a cohort of consistently ovary-specific genes

We observed several patterns in tissue-specific gene expression that are consistent across all twelve species. First, in all species the main axis of variation separated ovary RNA libraries from head and carcass (Fig D in S1 Text). This axis accounted for at least 50% of variation, and in several species greater than 70% of variation. To test for possible variation due to different runs on the sequencer, we resequenced several libraries and compared them using principle component analysis. We found variation between sequencing runs to be negligible compared to variation across tissues and individuals (Fig G in S1 Text). Second, in all species we observed that there was a larger amount of significantly downregulated transcripts than upregulated in the ovary relative to the carcass (Figs 2A and 2B, E in S1 Text). Across species, we observed an average of 25.4% to be significantly downregulated and 12.9% of transcripts to be significantly upregulated. When comparing the head to the carcass, we observed an average of 7% of transcripts to be significantly upregulated and 7.2% to be significantly downregulated (Fig F in S1 Text). These differences may reflect variation in the complexity and diversity of functions of the body parts being compared.

We used the results of our differential gene expression analysis within species to test for the existence of a suite of genes that show consistent ovary-specific expression across species. We found a cohort of 131 genes, grouped according to BLAST sequence similarity to *D. melanogaster*, for which at least one transcript was significantly upregulated in the ovaries of more than ten species (Fig 2C). Transcripts matching these genes made up on average 24.6% of the significantly ovary-upregulated transcripts across species, meaning roughly one quarter of ovary-specific genes have conserved expression patterns over evolutionary time. When excluding the species *S. varia*, this average decreased to 17.7%, as this species has the smallest set of ovary-upregulated transcripts, 100% of which correspond to core ovary genes.

We then tested the extent to which these core ovary genes correspond to observations in well-studied laboratory *Drosophila* models. To accomplish this, we compared expression across Hawaiian species to reported tissue-specific expression levels from *D. melanogaster* [29]. We found that Hawaiian core ovary-specific genes show nearly universal enrichment in the ovary of *D. melanogaster* as well, as reported in the FlyAtlas2 dataset [29] (Fig 2D). We

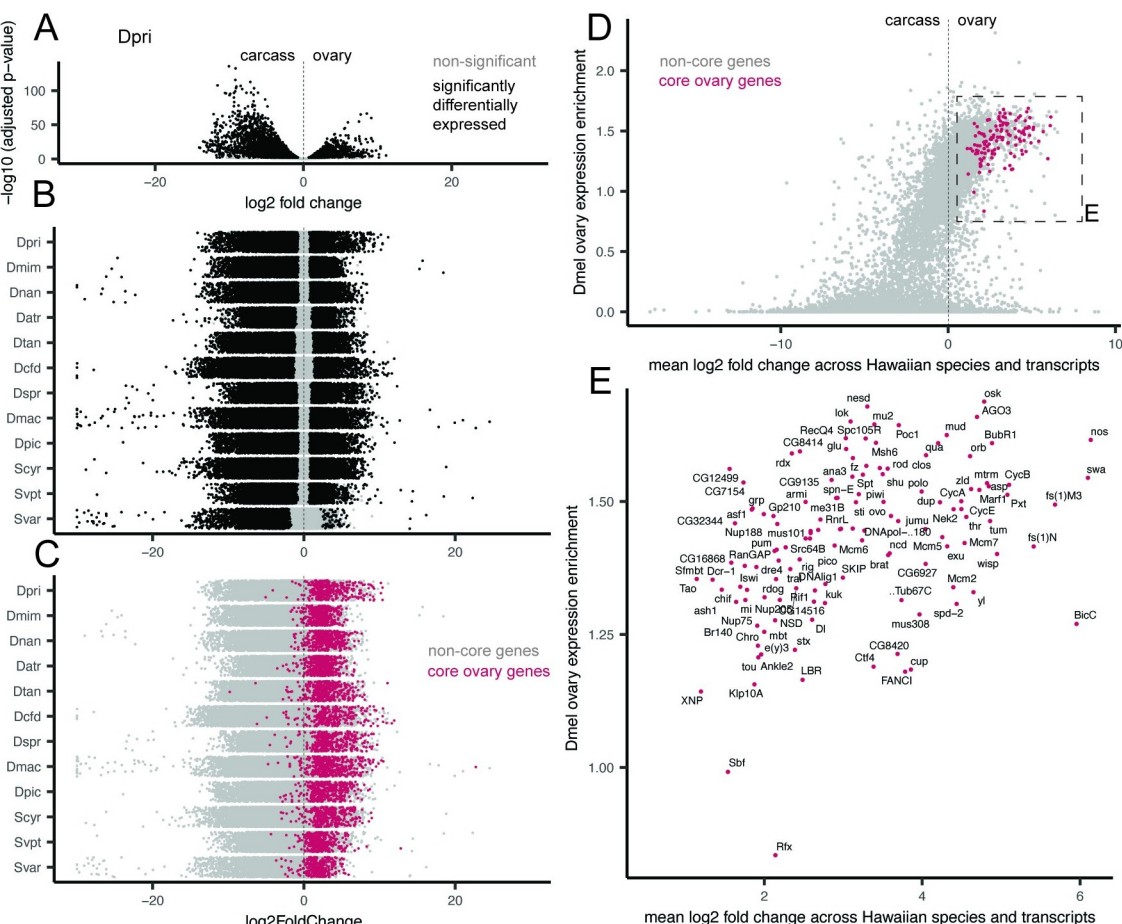

**Fig 2. Identifying a cohort of ovary-specific genes across drosophilid species.** A, Volcano plot for one example species, *D. primaeva* (Dpri), showing the results of a differential gene expression analysis comparing the ovary to the carcass. The x-axis shows the $\log_2$ fold change of expression across transcripts, and the y-axis shows the adjusted p-value, $\log_{10}$ transformed. Points that are significantly differentially expressed are shown in black. B, Jitter plots showing the results of the same analysis across the twelve species studied here. The x-axis shows the $\log_2$ fold change of expression across transcripts, and points are arranged with random jitter within species on the y-axis. C, The same jitter plots as in B, now colored according to whether or not transcripts belong to a cohort of core ovary genes. These are defined as genes, grouped by BLAST similarity to *D. melanogaster* transcripts, for which at least one transcript is upregulated in the ovary of ten or more of the twelve species. D, A comparison of mean expression change across Hawaiian species to reported ovary-enrichment values from *D. melanogaster*, as reported in FlyAtlas2 [29]. Core ovary genes are marked in magenta. E, The boxed region shown in D, magnified and now showing only core ovary genes, annotated with the gene symbol from *D. melanogaster*.

likewise observed that genes reported in *D. melanogaster* to have highest enrichment in the ovary largely correspond to genes that are significantly upregulated in the ovaries of Hawaiian species (Fig H in S1 Text).

The 131 core ovary genes include several well-known members involved in oogenesis and germline stem cell renewal such as *nanos* [34], *swallow* [35], and *oskar* [46](Fig 2E). We found only two genes that were identified as Hawaiian core ovary genes that are not reported in the FlyAtlas2 dataset [29] to be enriched in the ovary of *D. melanogaster*: the SET domain binding factor *sbf*, and *Rfx*, which are reported to be enriched in the heart, brain, and other non-reproductive tissues of *D. melanogaster* [29].

We used the same approach to identify a core suite of 52 head-specific genes (Fig I in S1 Text). There was no overlap between the sets of core head genes and core ovary genes. To test whether the correspondence between expression observations in Hawaiian flies and *D.*

*melanogaster* might be due to factors beyond tissue identity, we compared head expression values to ovary enrichment data from *D. melanogaster*, as we had done for ovary expression values above. We did not observe a correspondence in either direction between expression in the head of Hawaiian species and enrichment in the ovary of *D. melanogaster* (Fig J, panel a, in S1 Text). In contrast, we did find a correspondence between head-specific expression and genes enriched in the *D. melanogaster* brain, eye, and head (Fig J, panel b, in S1 Text). Core head genes include *Rhodopsin* photoreceptor genes and genes such as *hikaru genki* with involvement in synaptic centers [47].

## 2.2 Modeling reveals the phylogenetic decay of expression similarity between tissues

Many studies have investigated the question of whether we expect expression to be more similar across the same organ in different species, or across different organs within the same species [13–20]. Recent studies have suggested that the answer to this question will depend on the phylogenetic distance separating the species being compared [21]. Here we used a modeling approach to investigate this question with respect to the ovaries of Hawaiian drosophilids.

First, we determined an appropriate unit of comparison across species, based on an assessment of homologous features between reference transcriptomes. The agalma pipeline provides a method for determining homologous and orthologous sequences using an all-by-all BLAST approach to determine clusters of reciprocally similar sequences (homology groups). These can then be divided into orthology groups by estimating gene trees and identifying maximally inclusive subtrees with no more than one sequence per taxon [48]. We compared the representation of species across homology and orthology groups, and observed that while the representation of homology groups increases with the number of species compared, representation of orthology groups decreases (Fig K in S1 Text). This is a known obstacle in comparative transcriptomics, attributed to many transcripts being artifactually fragmented during reference transcriptome assembly [49]. To reduce the impact of this on our downstream analyses, we averaged TPM values across all transcripts within a homology group for each sequenced RNA library. Principle component analysis of this average expression dataset showed that the first principle component divides ovary libraries from the rest, while the second component separates samples along an axis that largely corresponds to phylogenetic distance between species (Fig L in S1 Text). While this averaging approach reduces noise due to variable mapping affinities of fragments of the same transcript, it comes at the cost of averaging over potential variation between genuine transcripts that fall into the same homology group. Future analyses using improved assemblies for transcriptomes or genomes will likely be able to avoid this trade off and compare transcript counts directly. To test the robustness of results to this averaging, we also performed key analyses over a dataset of the identifiable strict orthologs. In both datasets (based on homology groups or based on strict orthology groups), we averaged expression values across biological replicate individuals per species.

With average expression counts for homologous transcripts across species, we tested the degree to which variation across this dataset could be attributed to tissue-specific variation (here, ovary vs. carcass), species-specific variation, or neither (residual variation). Using the linear modeling approach adapted from Breschi and colleagues (2016) [21], we found the proportion of variance across the dataset attributed to tissue differences decreased with phylogenetic distance, while the proportion attributed to species difference increased (Fig 3A–3C). In addition, we found that, when comparing ovary and carcass tissues, the Hawaiian drosophilid clade encompasses the crossover point where variation across species swamps variation across tissues (crossed lines, Fig 3A). When comparing across the two species from the *picture-wing*

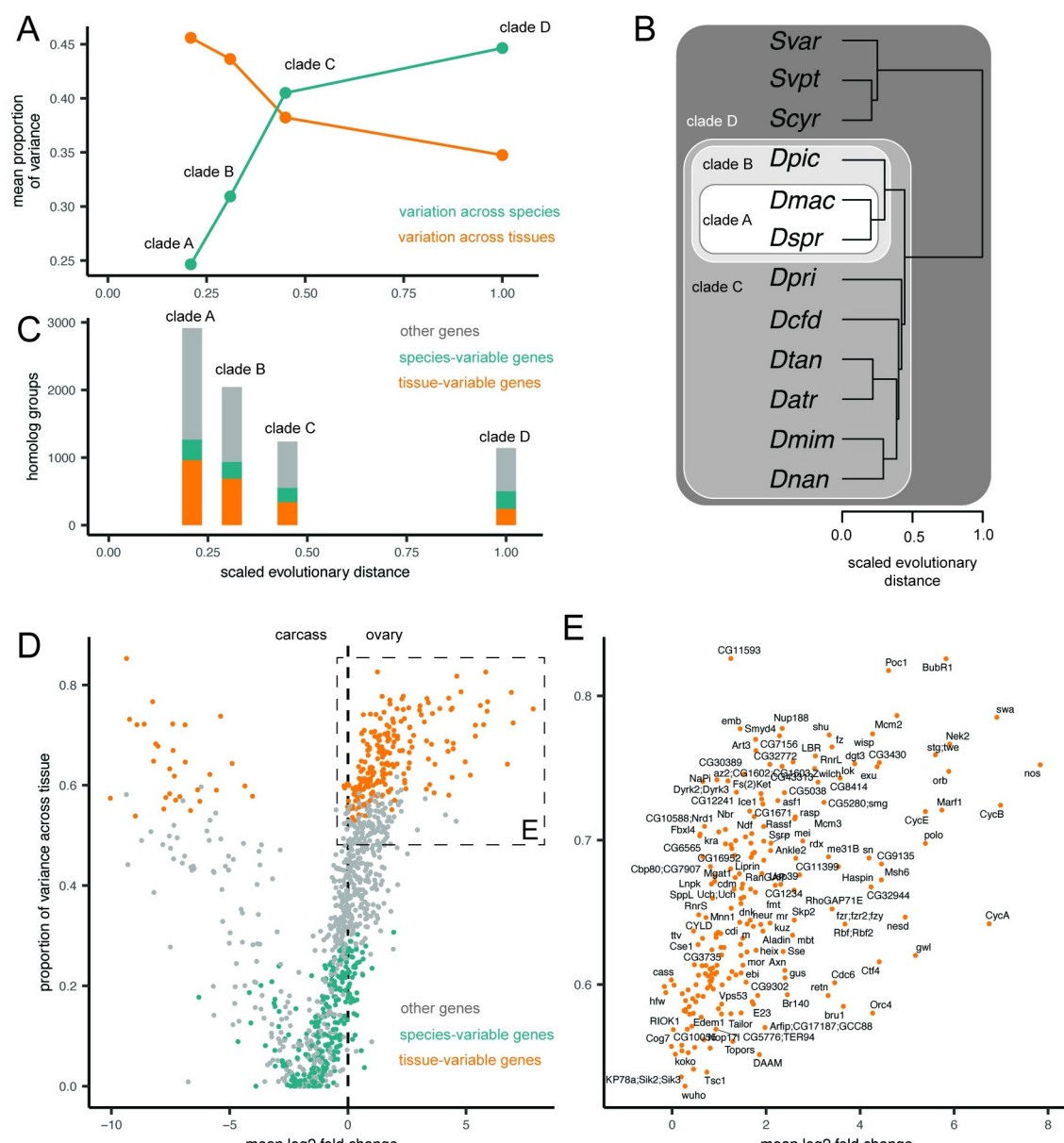

**Fig 3. Linear modeling shows the proportion of variance explained by differences across tissues and species.** A, The results of a linear modeling approach to calculate expression variation for each gene, attributed to variation across organs, species, or residual variation, as described in Breschi and colleagues (2016) [21]. The average proportion of variation attributed to tissues is higher than that attributed to species for the two *picture-wing* species in clade A, while the opposite is true for all twelve species in clade D. Sample sizes are for clade A: 2,918 homology groups, clade B: 2,044, clade C: 1,239, and clade D: 1,143. B, The number of genes, defined by homology group, classified as tissue variable genes (TVGs), species variable genes (SVGs), or neither in each clade comparison. C, The phylogeny of the twelve species studied here, showing the four clades compared in A-B. Scaled evolutionary distance is calculated as the relative distance from the most recent common ancestor of Hawaiian drosophilids to extant species. D, Comparing results of the differential gene expression approach ($\log_2$ fold change) on the x-axis to results of the modeling approach on the y-axis (variation across tissues). Genes are colored according to TVGs and SVGs. The inset box highlights TVGs that are upregulated in the ovary relative to the carcass. E, The same plot, now showing only upregulated TVGs, annotated with the gene symbol from the *D. melanogaster* sequences in the same homology group.

group included in this study, an average of 45.6% of the variation can be attributed to tissue differences. For the same comparison, 960 genes were identified as tissue-variable genes (TVGs), defined as residual variation accounting for <25% and a two-fold increase in variation attributed to tissues than to species (Fig 3B and M in S1 Text). In contrast, when comparing across all twelve Hawaiian drosophilid species studied here, 34.7% of the variation can be attributed to tissue, with 240 TVGs (Fig 3B and M in S1 Text). Across different clades of comparisons, the number of species-variable genes (SVGs) remains relatively stable (from 304 to 260, Fig 3B).

We then leveraged the results of this linear modeling approach across all twelve species to perform an additional screen for genes that are consistently upregulated in ovaries across species. We compared the proportion of variation explained by tissue for each homology group to the average $\log_2$ fold change from the results of our differential gene expression analysis (Fig 3D). This comparison allowed us to identify genes that fall above our threshold for TVGs that are also upregulated in the ovary (Fig 3E). This group of genes includes many of the same members as the core ovary genes (e.g. *nanos* and *swallow*), as well as several new candidates (e.g. *singed*).

To test the importance of tissue identity, we repeated the same analysis comparing variation across species and tissues using the head in place of the ovary. Consistent with what we describe for the ovary and carcass, as phylogenetic distance increases the proportion of variation across tissues decreases while variation across species increases. In contrast to the above findings, however, for the head and carcass far less of the variation in gene expression can be attributed to tissue differences (Fig N in S1 Text). For these tissues, the crossover point between total proportion of variation occurs roughly at the distance separating the two *picture-wing* species.

To verify these results were not driven by the species *S. varia*, which had the most distinct expression patterns of all species, we repeated these analyses excluding this species and recovered largely equivalent results (Fig O in S1 Text). To test robustness to homology group averaging, we repeated this analysis over strict orthologs, again recovering the same results (P in S1 Text). Finally, we also compared our findings to those that would be recovered using a more typical pairwise approach, by repeating the linear modeling analysis on ovary and carcass data using every pairwise combination of the twelve species. We recovered the same trend of decreasing contribution of tissue-level variation with increasing phylogenetic distance. While several pairs of species show more variation between species than tissues, we note that not every pair, nor the average across pairs, captures the crossover point where variation across species overwhelms variation across tissues (Fig Q in S1 Text). This reflects the inherent variability between individual pairwise comparisons of species, and highlights the importance of phylogenetic analytical approaches on entire clades.

## 2.3 Identifying gains and losses of ovary bias across genes and the phylogeny

While many ovary-specific transcripts belong to the cohort of core ovary genes, on average 75.4% of transcripts are upregulated in the ovaries of one or several species, but not consistently across ten or more of the species studied here (Fig 2B and 2C). This is suggestive of many evolutionary gains and losses of ovary-specific expression of genes. We characterized the evolution of these gains and losses using an ancestral state reconstruction approach. First we quantified expression bias between tissues as the ratio of read counts [7], then reconstructed the value of this continuous trait for each gene (defined using homology groups) at each node of the estimated species tree (Fig C in S1 Text). We then calculated the scaled change of

expression bias along each branch, which allowed us to describe how relative expression values between tissues had changed the course of evolutionary time (Fig 4A). Visualizing the distribution of scaled changes by genes shows that most scaled changes are small and centered around zero, representing little change in gene expression bias between tissues (Fig 4B–4C).

Using this dataset of scaled changes across genes and branches, we identified branches for which the direction of tissue bias had changed (e.g. from higher expression in the ovary than in the carcass to lower, or vice versa). Visualizing this dataset according to branches reveals that the majority of these changes in bias are located on the root and terminal branches, rather than internal branches (Fig 4D and 4E). This is likely because internal branches for this rapid radiation tend to be very short; even when scaling evolutionary changes to branch length, it is less probable for our analysis to identify a shift to and from ovary-biased expression on a short branch than a long branch. Repeating the same analysis on a dataset of strict orthologs reflects the same pattern, indicating that this result is not an artefact of expression averaging across homology groups (Fig R in S1 Text).

Visualizing the distribution of genes by ancestral and descendant values allows us to identify shifts in bias which represent the largest swings in expression values (Fig 4F, points a-d). Highlighting the top two such shifts in both directions, we identify four example genes which acquired or lost ovary-specific expression in the phylogeny of Hawaiian Drosophilidae. In the case of *FMRFaR* and *GABA*, a few Hawaiian species have gained ovary-biased expression of these genes, while most species and the ancestral state indicate non-ovary bias (Fig 4Fa-b). In the case of *vilya* and the unnamed gene *CG9109*, each shows a pattern where one species has lost ovary bias from a biased ancestral state (Fig 4Fc-d).

Repeating the same analysis using the head in place of the ovary revealed a set of evolutionary gains and losses in head-specific expression (Fig S in S1 Text). Identifying the top four changes in head expression shows gains and losses of head expression in the genes *hiro*, *stil*, *Jhe*, and, consistent with the ovary, *vilya*. In the case of the latter, these results may be driven by substantial changes in expression of *vilya* in carcass tissues across species, resulting in major differences in both ovary and head-biased expression.

## 2.4 Genes with a strong correlation of expression evolution

We tested the estimated evolutionary changes in expression bias for evidence of correlated expression evolution between genes. For every gene represented across all species, we performed a pairwise comparison of changes in expression bias, using as data points the scaled change in ovary bias on the 22 branches in the phylogenetic tree. This resulted in 1,306,449 pairwise measures of evolutionary correlation between genes. Because the number of gene pairs being compared is much larger than the number of values used to estimate correlation, this method has the potential to produce many spurious correlations [7]. To reduce the number of these correlations that are driven by low-confidence outliers [50], we removed any transcript with an estimated change in expression bias beyond two standard deviations from the mean change across all transcripts, and performed the same pairwise correlation analysis.

To test the degree to which the correlations observed here reflect known biological interactions between genes, we compared these measures to reported protein and genetic interactions between genes, using the database of published genetic experiments in *D. melanogaster*, available at http://flybase.org. We found that the mean correlation coefficient for genes that are known to physically interact as proteins was higher than for genes with no or unknown interaction (maximum p-value = <0.001 over 100 replicates, Fig 5A). This indicates that even with a relatively small number of observations, there is sufficient information in the matrix to detect biological signal between gene pairs. These results were calculated based on the correlation in

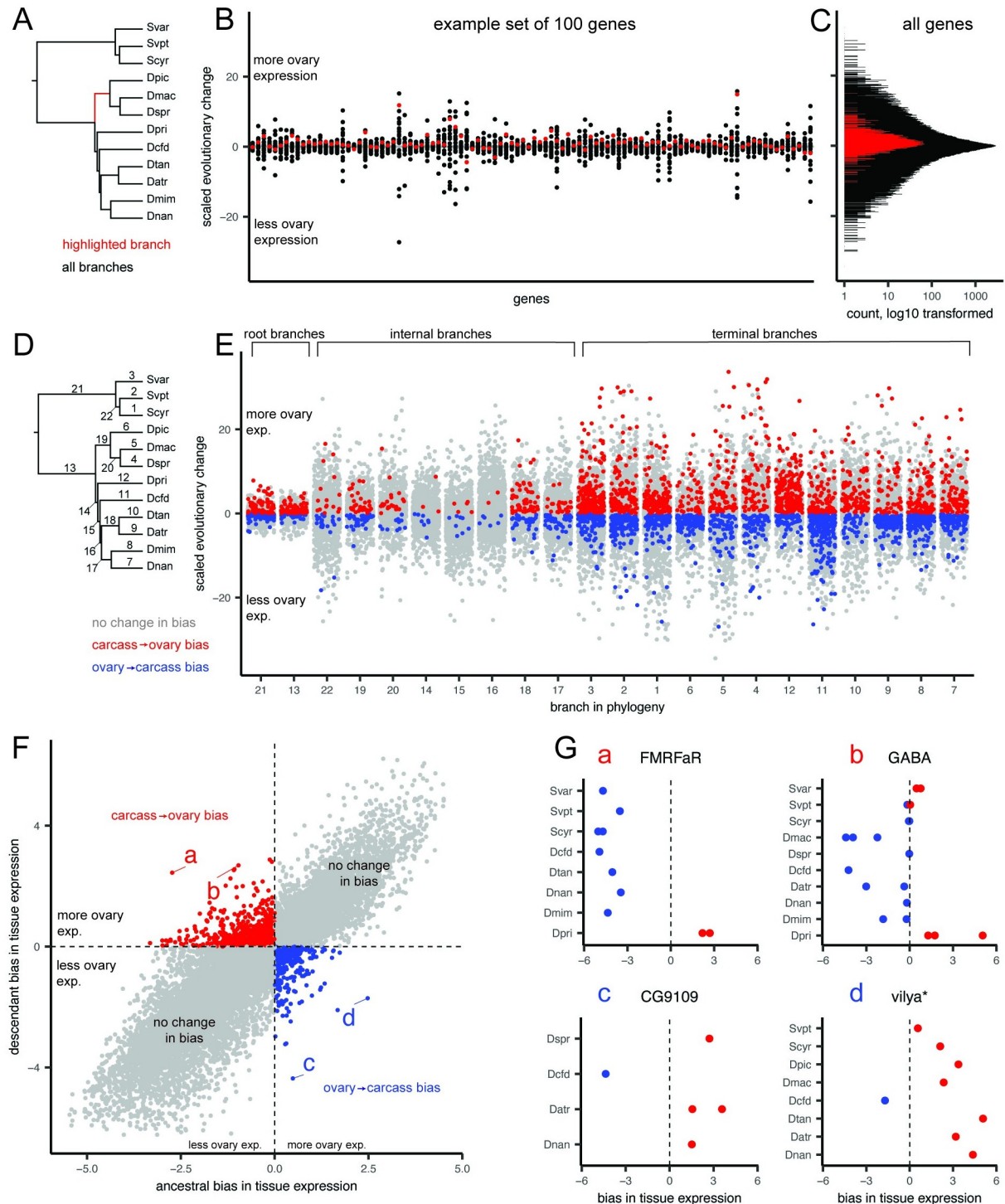

**Fig 4. Identifying genes that have gained and lost ovary-biased expression across the phylogeny.** A, The phylogeny of the twelve species studied here, highlighting one example branch of the 22 for which we inferred the scaled evolutionary change in expression bias. B, The distribution of changes, grouped by gene, for 100 randomly selected genes, defined by homology group. Each point represents one of the 22 branches from A, with the red point corresponding to the highlighted branch from that panel. C, The distribution, log$_{10}$ transformed, of scaled genes across all branches and all genes. Changes on the highlighted branch in red. D, The phylogeny with all 22 branches numbered. E, The distribution of changes, grouped by branch, with random jitter on the x-axis within each group. Points colored according to the qualitative change in bias, either from more expression in ovary than carcass to less (blue), the reverse (red), or no change in overall bias (gray). F, The distribution of ancestral and descendant values, showing the two quadrants that represent qualitative changes in bias. Points that represent large

swings in expression within those quadrants are labeled a-d. G, The four genes with large swings from F, showing the expression bias for each transcript colored according to more expression in the ovary (red) or carcass (blue). Panels annotated with the gene symbol from the *D. melanogaster* sequences in the same homology group, with the exception of *vilya**, which was annotated using a direct BLAST search since no *D. melanogaster* sequence was present in that group.

expression bias between the ovary and carcass. However, following the same procedure using correlations in changes in head-biased expression showed no significant difference between the two groups (max. p-value = 0.256, Fig T in S1 Text), suggesting the strength of this signal may be dependent on the tissues being compared.

We also found that genes known to interact genetically have a significantly higher mean correlation than genes with no or unknown genetic interactions (unknown vs. enhancement

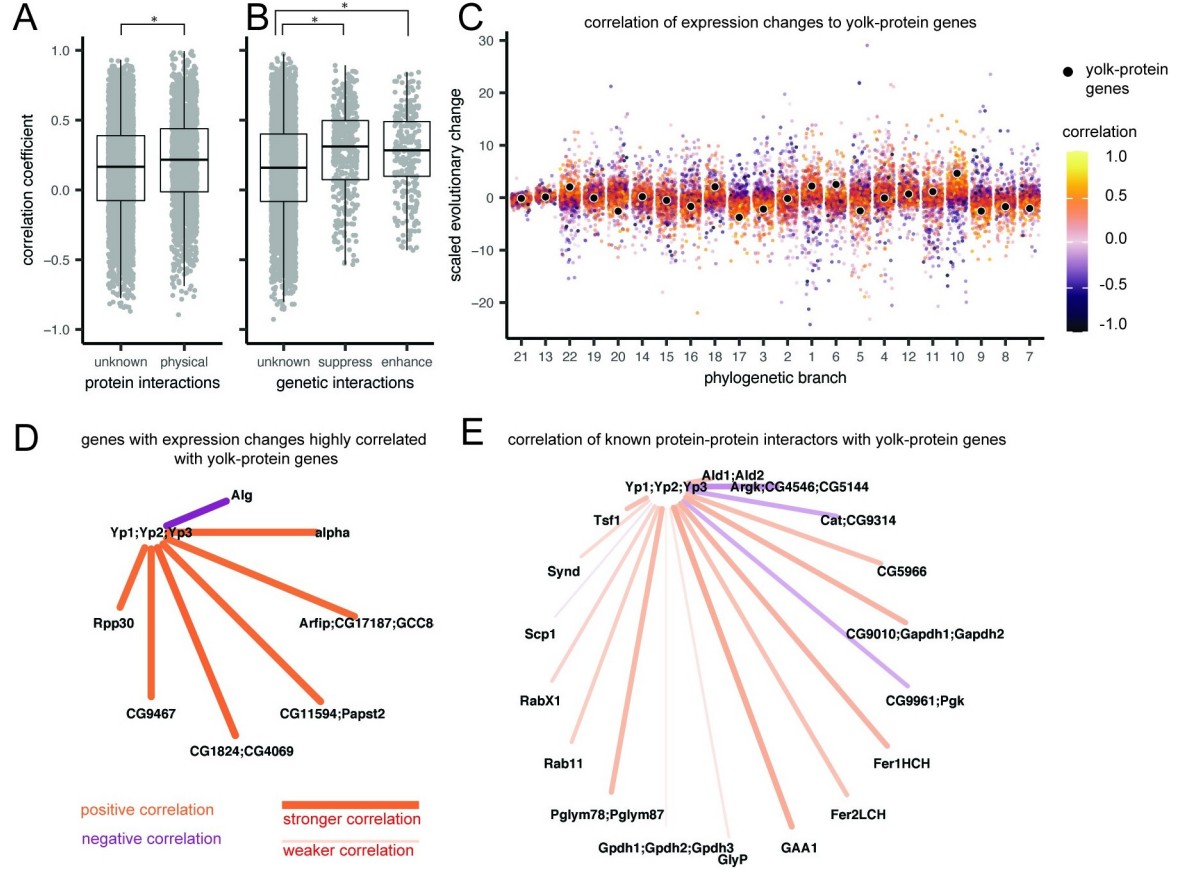

**Fig 5. Estimating pairwise correlation coefficients across genes reveals new networks of correlated expression evolution.** A-B Comparison of the distribution of Pearson's correlation coefficients based on ovary-biased expression evolution between genes. Box plots indicate mean, upper and lower quartiles, and 1.5x interquartile ranges. Asterisks indicate a significant t-test comparison. A, Genes with no or unknown protein-protein interactions compared to those with reported interactions in FlyBase [52] (maximum p-value = <0.001 over 100 replicates). B, Correlation comparison between genes with no or unknown genetic interactions and those reported to have enhancement or suppression interactions in FlyBase (unknown vs. enhancement max. p-value = <0.001; unknown vs. suppression max. p-value = <0.001; enhancement vs. suppression p-value = 0.497). C, Each point represents a scaled change in expression bias, colored by Pearson's correlation coefficients relative to one example gene-family, the yolk-protein genes (black points), arranged by phylogenetic branch (numbers shown in Fig 4D). Yellow = strong positive correlation, purple = strong negative correlation. D, The network of strong correlation partners (absolute correlation > 0.825) with the yolk-protein genes, colored by the direction of correlation. Stronger correlations are shown by brighter colors, and thicker, shorter lines. Nodes are annotated with the gene symbols from the *D. melanogaster* sequences from that homology group. E, The correlation between known protein-protein interaction partners [52] with the yolk-protein genes.

max. p-value <0.001, unknown vs. suppression max. p-value = <0.001, Fig 5B). Comparing genes with known genetic enhancement and suppression interactions to each other showed no significant difference (p-value = 0.497). However, for genetic interactions, the range of correlation coefficients was higher in the group of no or unknown interactions (Fig 5B). This indicates that, while the average correlation of expression evolution might be higher for interaction partners, stronger positive and negative correlations exist between pairs of genes which do not interact, or for which interactions have not yet been tested. We observed the same differences in correlations after removing low-confidence outliers, though support for the significant difference between physically interacting proteins was reduced (7 of 100 iterations had a p-value >0.001, the maximum p-value was 0.007).

We tested whether the network inferred based on strong correlation of expression evolution was consistent with known interaction partners from *D. melanogaster*. We selected as an example the gene yolk-protein gene family, which are known to be expressed in the reproductive system, among other tissues [51] (Fig 5C). We found eight distinct homologous gene groups, comprising 12 unique *D. melanogaster* parent genes, that had a strong evolutionary correlation with yolk-protein genes (absolute coefficient greater than 0.825, Fig 5D). None of these correlated genes correspond to those listed on FlyBase [52] as having known interactions with yolk-protein genes in *D. melanogaster* (Fig 5E). We consider these strong evolutionary correlations to be a set of new predictions about evolutionary and genetic relationships between genes which can be tested in wild and laboratory model species of *Drosophila*. The correlation networks for each gene family, estimated using the full dataset or using the more conservative dataset with low-confidence outliers removed, can be interrogated in the accompanying data visualization for this manuscript (https://shchurch.shinyapps.io/hawaiian_fly_dataviz_2021).

## 3 Discussion

The results of this study show the importance of placing any comparison of gene expression across species in an evolutionary context. When making comparisons that involve model organsims for the study of development and disease, this means identifying the crossover point at which variation between species begins to swamp variation across the tissues or treatments in question. In such comparisons, the possibility that any individual gene may show a divergent pattern of expression from the model organism increases substantially. This study provides evidence that confirms we should expect variation in gene expression to increase with the phylogenetic distance separating the species being compared. In addition, our results using ovary and head expression data show that our expectation should also depend on the identity of the tissues being compared. For some tissues, like the fly head, this crossover point may be met even when comparing between two relatively closely related species. In summary, consistent with recent findings in mammals [21], our study shows that our expectation about the interspecific similarity of tissues should decrease as the phylogenetic distance increases, and the rate of that decrease will depend on the tissue in question.

Despite substantial variation across species, we identified core suites of ovary- and head-expressed genes that have maintained conservation of expression patterns over millions of years of evolution. The core ovary genes include some of the most well-studied genes in relation to *D. melanogaster* oogenesis, such as *nanos* and *oskar*, as well as many genes that have yet to be studied in depth (e.g. unnamed genes such as *CG3430*). We provide the full list of core ovary and head genes as a reference against which future genetic studies may be informed and compared (Tables F-G in S1 Text). Furthermore, the existence of these suites of genes suggests that equivalent groups are likely to exist within the many gene expression atlases currently

being published [53, 54]. New technologies such as single-cell RNA sequencing that use global signatures of gene expression to identify cells are ripe for interspecific comparisons that may reveal evolutionarily conserved gene modules [55]. Developing robust comparative methods for comparing these atlases across species has the potential to reveal ancestral expression patterns in cells and organs, as well as pinpoint important evolutionary shifts in expression regulation.

Our results indicate that genes known to interact, both physically as the proteins they encode and through genetic enhancement and suppression, likely experience more correlated changes in expression than would be expected for genes chosen at random. However, we also find the difference in mean correlation between these groups to be relatively small, and dependent on the context of the tissue in question. One possible explanation for this finding is that interactions between genes with strong correlations of expression evolution have yet to be described. We provide an interactive tool to explore highly correlated genes that can inform future genetic studies in *D. melanogaster* and other related species (https://shchurch. shinyapps.io/hawaiian_fly_dataviz_2021). Another possibility we consider likely is that interactions between genes represent only one factor among many that dictate the probability of correlated changes in expression. We hypothesize that other features, such as shared regulatory or chromatin architecture, will also influence evolutionary correlation of expression.

As more studies undertake phylogenetic comparisons of functional genomic data, new factors that influence the evolutionary associations between genes are likely to be revealed [7]. The strength of these phylogenetic comparisons will depend in part on comparing across a sufficient number of taxa such that there are multiple branches on which to calculate and compare evolutionary changes. However, even as functional genomic data become more accessible for more species, the number of features being compared (e.g. thousands of genes) will likely continue to outnumber the number of evolutionary observations (e.g. changes along branches) [7]. One encouraging result from this study is that, using our matrix of gene expression changes along 22 branches, we find sufficient information to detect the biological signal associated with physical and genetic interactions. While this is true, we assume that some fraction of the correlations that we report here represent false positives, and that the strength of correlation of these genes would decrease with the addition of more taxa to the comparison. For this reason we present the correlation matrix as a set of hypotheses to be tested in future studies using additional lines of evidence.

One outstanding challenge in expression evolution is the quality of the references available against which RNA reads can be mapped [49]. In this study we account for variation due to assembly artifacts by averaging expression values over groups of homologous genes, as identified by sequence similarity to high quality reference genomes. This approach has the advantage of accounting for problems associated with fragmentation of genes in transcriptome assembly. However, it comes at the cost of averaging over possible biological variation in expression between genes from the same gene family. The concordance of our results with published findings from *D. melanogaster* suggests that the approach we have used here is robust for our dataset. However, as the quality and accessibility of genomes from diverse species continue to increase, future studies will likely be able to compare directly between orthologous genes without needing to account for fragmentation. For those future studies, a phylogenetic comparative approach like the one used here and elsewhere [8] can serve as an analytical framework to move expression comparisons beyond pairwise comparisons.

One goal of evolutionary developmental biology is to identify changes in developmental mechanisms that underlie phenotypic differences [12]. Many studies approach this by identifying phenotypic variation between species and then searching for differences in gene content or expression using one or several emerging model organisms in the lab [12]. To narrow down

the field of search, this approach often requires *a priori* knowledge of candidate genes, gained from developmental research in related models or other methods of filtering the genome. Furthermore, because these approaches usually lack global measurements of gene expression variation across species, identifying an expression difference does not always constitute a smoking gun [6]. For example, observing a difference in candidate gene expression between taxa would not be unexpected if we frequently observe differences of that magnitude between genes chosen at random. An alternative approach, as demonstrated here, is to characterize all the evolutionary changes in expression across the transcriptome, and then identify the changes that are significantly associated with traits of interest [9]. As expression data become available from an ever wider array of species, this "evolutionary screen" approach becomes increasingly possible. One advantage of this approach is that it may reveal associations that would otherwise escape detection when comparisons are centered on model organisms; for example, when genes, traits, or processes happen to not be present in our laboratory model species [10]. By leveraging phylogenetic comparative methods on high-dimensional functional genomic data, the objective of connecting genomic variation to developmental mechanisms and phenotypic differences will be accelerated.

## 4 Methods

### 4.1 Field collection

Specimens used for transcriptome sampling were caught on the Hawaiian islands between May of 2016 and May of 2017. Specimens were caught using a combination of net sweeping and fermented banana-mushroom baits in various field sites on the Hawaiian islands of Kaua'i and Hawai'i (see Table A in S1 Text for locality data). Field collections were performed under permits issued by the following: Hawai'i Department of Land and Natural Resources, Hawai'i Island Forest Reserves, Kaua'i Island Forest Reserves, Koke'e State Park, and Hawai'i Volcanoes National Park. Adult flies were maintained in the field on vials with sugar media and kept at cool temperatures. They were transported alive back to Cambridge, Massachusetts where they were maintained on standard *Drosophila* media at 18˚C. Samples were processed for RNA extraction between 5 and 31 days after collecting them live in the field (average 10.8 days, see Table A in S1 Text). One species, *Scaptomyza varia*, was caught in the field before the adult stage by sampling rotting *Clermontia sp.* flowers (the oviposition substrate). For this species, male and female adult flies emerged in the lab, and were kept together until sampled for RNA extraction.

### 4.2 Species identification

Species were identified using dichotomous keys [56–60], when possible. Many keys for Hawaiian Drosophilidae are written focusing on male specific characters (e.g. sexually dimorphic features or male genitalia) [58]. Therefore, for species where females could not be unambiguously identified by morphology, we verified their identity using DNA barcoding. When males were caught from the same location, we identified males to species using dichotomous keys and matched their barcode sequences to females included in our study. We also matched barcodes from collected females to sequences previously uploaded to NCBI [61–63].

The following dichotomous keys were used to identify species: for *picture-wing* males and females, Magnacca and Price (2012) [56]; for *antopocerus* males, Hardy (1977) [57]; for *Scaptomyza*, Hackman (1959) [58]; for species in the *mimica* subgroup of MM, O'Grady and colleagues (2003) [59]; for other miscellaneous species, Hardy (1965) [60].

For DNA barcoding, DNA was extracted from one or two legs from male specimens using the Qiagen DNeasy blood and tissue extraction kit, or from the DNA of females isolated

during RNA extraction (see below). We amplified and sequenced the cytochrome oxidase I (COI), II (COII) and 16S rRNA genes using the primers and protocols described in Sarikaya and colleagues (2019) [40].

For barcode matching, we aligned sequences using MAFFT, version v7.475 [64], and assembled gene trees using RAxML, version 8.2.9 [65]. Definitive matches were considered when sequences for females formed a monophyletic clade with reference males or reference sequences from NCBI; see Table B in S1 Text.

Female *D. primaeva*, *D. macrothrix*, *D. sproati*, and *D. picticornis* could be identified unambiguously using dichotomous keys. Female *D. atroscutellata*, *D. nanella*, *D. mimica*, *D. tanythrix*, *S. cyrtandrae*, *S. varipicta*, and *S. varia* were identified by matching barcodes to reference sequences from NCBI, reference males, or both. For the female *haleakalae* fly used in this study, no male flies were caught in the same location as these individuals, and no other sequences for *haleakalae* males on NCBI were an exact match with this species. Given its similar appearance to *Drosophila dives*, we are referring to it here as *Drosophila* cf *dives*, and we await further molecular and taxonomic studies of this group that will resolve its identity.

### 4.3 Sampling strategy

The target number of mature, healthy female flies per species was four, with three intended for dissection and species-specific expression libraries and one intended as a whole-body reference library (Fig 1). When four such individuals were not available, a reference library was assembled by combining the tissue-specific libraries from one of the other individuals. This was the case for the following species: *D. sproati*, which was dissected and had RNA extracted separately from the head, ovaries, and carcass, with RNA combined prior to library preparation; and *S. varia*, *S. cyrtandrae* and *D.* cf *dives*, for which RNA was extracted and libraries prepared for separate tissues, and raw reads were combined after sequencing.

For the other eight species, sufficient individual females were available such that reads for transcriptome assembly were sequenced from a separate individual. In these cases one entire female fly was dissected and photographed to assess whether vitellogenic eggs were present in the ovary, and all tissues were combined in the same tube and used for RNA extraction. Library preparation failed for one individual *D. atroscutellata* fly, as well as two tissue-specific libraries: one head sample from *D. mimica*, and one head sample from *D. sproati*.

### 4.4 Dissection and RNA sequencing

Female flies were anesthetized in 100% ethanol and were dissected in a 1x phosphate-buffered saline solution. The ovary was separated from the abdomen, and the head was separated from the carcass. Photographs of each tissue were taken, and tissues were moved to pre-frozen eppendorf tubes, kept in dry ice, and immediately transported to a -80˚C freezer. Dissections were performed as quickly as possible to prevent RNA degradation. Samples were stored at -80˚C for between 90 and 336 days before RNA extraction (average 281.9 days, see Table A in S1 Text).

RNA was extracted from frozen samples using the standard TRIzol protocol (http://tools.thermofisher.com/content/sfs/manuals/trizol_reagent.pdf). One mL of TRIzol was added to each frozen sample, which were then homogenized using a sterile motorized mortar. The recommended protocol was followed without modifications, using 10 $\mu$g of glycogen, and resuspending in 20$\mu$L RNAse-free water-EDTA-SDS solution. DNA for subsequent barcoding was also extracted using the phenol-chloroform phase saved from the RNA extraction.

RNA concentration was checked using a Qubit fluorometer, and integrity was assessed with a Agilent TapeStation 4200. RNA libraries were prepared following the PrepX polyA mRNA

Isolation kit and the PrepX RNA-Seq for Illumina Library kit, using the 48 sample protocol on an Apollo 324 liquid handling robot in the Harvard University Bauer Core Facilities. Final library concentration and integrity were again assessed using the QUbit and TapeStation protocols.

Samples intended for transcriptome assembly were sequenced on an Illumina HiSeq 2500, using the standard version 4 protocol, at 125 base pairs of paired-end reads. Samples intended for tissue-specific expression analyses were sequenced on an Illumina NextSeq 500, using a high output flow cell, at 75 base pairs of single-end reads. A table of total read counts for each library can be found in Tables C-D in S1 Text. To account for any possible batch effects across separate rounds of sequencing, each sequencing run was performed with one or several over-lapping samples. Principle component analysis of these libraries showed variation between sequencing runs to be negligible relative to variation between tissue and individual (see Results and Fig G in S1 Text).

### 4.5 Transcriptome assembling and expression mapping

Transcriptome assembly and expression mapping was performed using the agalma pipeline, version 2.0.0 [48]. For the twelve reference transcriptomes, reads from separate rounds of sequencing were concatenated and inserted into the agalma catalog. Further details of tran-scriptome assembly and homology assessment are included in our previous manuscript [44].

Each tissue-specific expression library was mapped to the corresponding reference tran-scriptome using the 'expression' pipeline in agalma, which uses the software RSEM to estimate gene and isoform count levels from RNAseq data [66]. The agalma pipeline also includes steps to catalog the species, tissue type, and run information, which were exported as a single JavaScript object notation (JSON) file. This file is available in the GitHub repository in the directory `analysis/data`.

### 4.6 Phylogenetic analysis

The methods for inferring a phylogenetic tree of the species here are the same as those described in our previous manuscript [44]. In summary, we used the agalma pipeline [48] to infer homology, based on an all-by-all BLAST procedure. Gene orthology was inferred based on the topology of gene trees, estimated with RAxML [65], version 8.2.9. The most likely tree was inferred using IQtree [67], version 2.1.1, on a dataset partitioned by transcripts and using the default Model Finder [68]. For this study, gene trees were additionally annotated with the software phyldog [69].

### 4.7 Annotating transcripts by sequence similarity

We leveraged the close relationship of these species to species of *Drosophila* with well-anno-tated genomes to annotate the transcripts considered here. For each transcript in the reference transcriptome, we performed four comparisons of sequence similarity using local BLAST: [1] comparing nucleotide transcript sequences to nucleotide sequences from *D. melanogaster* (blastn), [2] comparing translated nucleotide sequences to protein sequences of *D. melanoag-ster* (blastx), [3] comparing nucleotide sequences to a database of nucleotide sequences from *D. melanogaster*, *D. virilis*, and *D. grimshawi* (blastx), and [4] comparing translated nucleotide sequences to a database of protein sequences from the same three species (blastn). For down-stream analyses, we prioritized annotations from the second comparison, but we provide all sequence similarity reports in the GitHub repository under the directory `analysis/BLAST`.

To annotate homology groups as defined by the homology inference step of agalma, we extracted the name and sequence ID from all *D. melanogaster* sequences in the group.

## 4.8 Normalization and differential gene expression

Transcript count tables were imported into R using the agalmar package, version 0.0.0.9000. Differential gene expression analysis was performed using the package DESeq2, version 1.34.0. For these analyses we used only one sequencing run per library, thereby excluding duplicate sequencing runs. Analyses of differential gene expression were calculated using the default approaches in DESeq2 for estimating size factors, dispersions, and calculating $\log_2$ fold-change and adjusted p-values using the Benjamini and Hochberg correction (Fig B, panel a, in S1 Text). Both individual and tissue were considered in the design formula. Transcripts were considered differentially expressed at a significance threshold of 0.01.

We identified a cohort of core ovary-specific genes by first identifying a parent gene for each transcript using a sequence similarity search against *D. melanogaster* (Fig B, panel a, in S1 Text). We then identified parent genes that had at least one transcript significantly differentially upregulated in the ovary of more than ten of the twelve species. Because multiple transcripts may match to a single parent-gene, core ovary-specific parent genes may include transcripts that are also not differentially upregulated in the ovary, as long as at least one transcript is upregulated for more than ten out of twelve species. This may be the case when transcripts are artificially fragmented during reference transcriptome assembly, or when sequence-similar transcripts have biologically distinct expression levels.

## 4.9 Comparison of expression to *D. melanogaster*

We compared our differential gene expression results to a reference database of tissue expression from *D. melanogaster*, known as the FlyAtlas2 [29]. We downloaded this reference in July of 2021, from http://motif.gla.ac.uk/downloads/FlyAtlas2_21.04.18.sql. This dataset provides data on transcript abundance and tissue enrichment, including for female ovaries. Tissue enrichment is calculated using the same methods as in the FlyAtlas2 web browser, defined as the fragments per kilobase of transcript per million mapped reads (FPKM) for a given tissue divided by that value for the reference tissue (here, female whole body), with a pseudocount of two counts added to empty values to avoid division by zero. We considered a FlyAtlas gene to be enriched in the ovary, comparable to our data, if the ovary was the maximum enrichment value across all tissues excluding the head, brain, and eye tissues, as these were separated in our RNASeq procedure (Fig B, panel a, in S1 Text). We considered a FlyAtlas gene to be head enriched if either the head, brain, or eye were the maximum enrichment value, excluding the ovary.

## 4.10 Transforming data into comparable measurements of expression across species

Transcript counts are reported in transcripts per million (TPM), but this measurement is known to not be directly comparable across species due to differences in reference transcriptome size [7, 8]. Therefore, we normalized TPM by species using the procedure described by Munro and colleagues (2021) [8], where TPM values are multiplied by the number of genes in the reference, and this value is divided by $10^4$ (Fig B, panel b, in S1 Text). TPM10k values were natural-log transformed.

An additional challenge when working with reference transcriptomes is the presence of fragmented transcripts created during the assembly process [49]. This fragmentation can result in noise in estimating the amount of transcript as reads are differentially mapped to these

fragments. To reduce the impact of this noise on our analysis, we undertook a novel approach where transcripts were grouped according to inferred homology as estimated by the agalma pipeline using an all-by-all BLAST approach (Fig B, panel b, in S1 Text). For each sequenced library, we found the average count value across all transcripts from the same homology group (see Table E in S1 Text for statistics on homology group composition). For each species-tissue pair, we then averaged this value across all biological replicates, here replicate individuals.

### 4.11 Linear modeling

We performed linear modeling to calculate the relative contribution of tissue- and species-level differences to variation in gene expression (Fig B, panel b, in S1 Text), following the approach of Breschi and colleagues (2016) [21]. These analyses were performed separately on datasets of ovary vs. carcass and head vs. carcass expression. Using the ANOVA script provided at https://github.com/abreschi/Rscripts/blob/master/anova.R, we built a linear model for each gene that accounts for the contribution of the organ, species, and any residual error. We then calculated the relative proportion of each factor divided by the total sum of squares for all factors. We identified groups of highly variable genes, using the same metrics defined by Breschi and colleagues (2016) [21], as any gene for which either tissues or species explains at least 75% of the variance. Species variable genes (SVGs) were defined as highly variable genes whose relative variation was two-fold greater across species than tissues (vice-versa for tissue variable genes, TVGs).

We performed these linear model analyses over four nested clades: a clade containing two *picture-wing* species (*D. sproati* and *D. macrothrix*); a clade containing the four *picture-wing-Nudidrosophila-Ateledrosophila* species in this study; a clade containing the nine Hawaiian *Drosophila* species in this study; and a clade of all 12 Hawaiian *Drosophila* and *Scaptomyza* species in this study. We repeated these analyses excluding the species *S. varia*, which showed the lowest similarity in expression to the other eleven species. To compare our analysis to the more typical approach, we also performed these analyses on all pairwise combinations of these twelve species.

### 4.12 Reconstructing evolutionary history of differential expression

We calculated tissue bias as the ratio of counts in TPM10k for each tissue (ovary and head) to the reference tissue [7], here the carcass (Fig C, panel a, in S1 Text). Using ratios is advantageous because any potential variation in transcript counting efficiency across individuals is canceled out, as this will be present in both the numerator and denominator [7]. We subsequently performed the same transformation steps described above, averaging over ratios from the same homology group and across biological replicates, to calculate average expression bias per homology group per library. To avoid division by zero, we added a pseudocount of 0.01 to each TPM10k value. Ratio values were natural-log transformed so that positive values indicate enrichment in the tissue of interest relative to the reference tissue, negative values indicate the opposite, and values of zero indicate equivalent expression.

We reconstructed the evolutionary history of tissue bias for each homology group using the species tree published in Church and Extavour, 2021 [44], based on the same reference transcriptome data (Fig B, panel c, in S1 Text). First, we calibrated the tree estimated using IQtree (Fig 1A of that publication) to be ultrametric using the R function `chronos` in the package `ape`, version 5.6.2 (using a correlated model and a lambda value of 1). We then subset this tree to only include tips for which expression data was available, and annotated this tree to be able to identify specific branches and nodes in ancestral state reconstruction analyses.

Ancestral expression bias values were estimated with the R package Rphylopars, version 0.3.9, using the fast ancestral state reconstruction algorithm based on Ho and Ané, 2014 [70] (Fig C, panel a, in S1 Text). Tips for which expression data were not available were dropped from each reconstruction, and ancestral state reconstruction was only performed when more than three tips had data. Following ancestral state reconstruction, we calculated the scaled change as the difference between the value at the ancestral and descendant nodes, divided by the length of the branch. Scaled changes were compared between homology groups by identifying equivalent branches as those that share the same parent and child node, following the procedure described in Munro and colleagues (2021) [8]. We identified qualitative changes in expression bias as changes that resulted in a ratio changing from negative to positive values or vice versa.

### 4.13 Estimating correlated evolution of expression across genes

For each homology group that had representation across all twelve species, we calculated pairwise Pearson's correlation coefficients by comparing scaled changes in expression bias across equivalent branches (Fig C, panel b, in S1 Text). For the twelve-species phylogeny, this meant each correlation coefficient was calculated using 22 individual data points (branches). This resulted in a correlation matrix of 1,306,449 pairwise comparisons of evolutionary correlation.

We compared this correlation network to data on protein interactions and genetic interactions downloaded from http://flybase.org in July, 2021. These data include pairwise observations of genetic enhancement and suppression interactions between parent genes in *D. melanogaster*. These interactions were matched to pairwise correlation coefficients by identifying the corresponding homology group for each *D. melanogaster* parent gene ID (more than one parent gene may fall into the same homology group).

We tested whether correlation coefficients for known physical and genetic interaction partners were higher than in genes with unknown interactions using two-sample t-tests. The sample size for physical interaction partners was 1,953, for genetic enhancement was 280, and for genetic supression was 497. In each test we compared the coefficients for either enhancement or suppression interactions to a random sample of 5000 coefficients for which interactions are unknown. We repeated these t-tests 100 times using different random samples, and report the maximum p-value observed. We also compared the distribution of enhancement and suppression interaction coefficients to each other using a single t-test.

Strong correlations for the visualization of co-evolutionary networks were selected using a threshold correlation coefficient of 0.825. Because spurious correlations may be driven by the presence of low-confidence outliers [50], we repeated all correlation analyses after removing any transcript with an estimated change in expression bias beyond two standard deviations from the mean change across all transcripts.

## Supporting information

**S1 Text. Supplementary Methods and Tables.** Additional figures and text describing analysis methods, as well as tables of sequencing statistics and results.
(PDF)

## Acknowledgments

We thank Didem Sarikaya, Karl Magnacca, and Steve Montgomery for their field assistance and expertise in Hawaiian fly identification and husbandry. We thank Kenneth Kaneshiro for

the use of his lab space in preparing wild caught specimens. We thank Bruno de Medeiros, Seth Donoughe, and members of the Dunn and Extavour labs for discussion of ideas.

## Author Contributions

**Conceptualization:** Samuel H. Church, Cassandra G. Extavour.

**Data curation:** Samuel H. Church.

**Formal analysis:** Samuel H. Church.

**Funding acquisition:** Samuel H. Church, Cassandra G. Extavour.

**Investigation:** Samuel H. Church.

**Methodology:** Samuel H. Church, Catriona Munro, Casey W. Dunn.

**Project administration:** Cassandra G. Extavour.

**Resources:** Cassandra G. Extavour.

**Software:** Samuel H. Church, Catriona Munro, Casey W. Dunn.

**Supervision:** Casey W. Dunn, Cassandra G. Extavour.

**Validation:** Samuel H. Church.

**Visualization:** Samuel H. Church.

**Writing – original draft:** Samuel H. Church.

**Writing – review & editing:** Samuel H. Church, Catriona Munro, Casey W. Dunn, Cassandra G. Extavour.

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
