## [Decision Letter · Decision Letter 0]

7 Sep 2022

Dear Dr Church,

Thank you very much for submitting your Research Article entitled 'The evolution of ovary-biased gene expression in Hawaiian Drosophila' to PLOS Genetics.

The manuscript was fully evaluated at the editorial level and by independent peer reviewers. The reviewers appreciated the attention to an important problem, but raised some substantial concerns about the current manuscript. Reviewer #2, in particular, raises some questions about the robustness of statistical inferences based on pairwise correlations, and suggests potential alternative approaches that you may wish to consider. Confirming the predictive value of the co-evolutionary analysis (e.g., in the inference of "new networks of genes that share evolutionary trajectories") by validation experiments, as that reviewer suggests, would of course be great, but I think that goes beyond the scope of this paper.  Demonstrating the robustness of the co-evolutionary analysis itself seems more important to this study, specially given the small magnitude of some of the differences you observe.  I suggest paying particular attention to the major point 1 and minor points 1 and 4 of the 2nd reviewer.  As for the "why reconstruct bias" question raised by Reviewer #3, I believe that has been discussed in Casey's previous papers, but a more thorough recap of that logic would help keep the reviewers and readers on board.

Based on the reviews, we will not be able to accept this version of the manuscript, but we would be willing to review a much-revised version. We cannot, of course, promise publication at that time.

If you decide to revise the manuscript for further consideration at PLOS Genetics, please aim to resubmit within the next 60 days, unless it will take extra time to address the concerns of the reviewers, in which case we would appreciate an expected resubmission date by email to plosgenetics@plos.org.

[LINK]

We are sorry that we cannot be more positive about your manuscript at this stage. Please do not hesitate to contact us if you have any concerns or questions.

Yours sincerely,

Artyom Kopp

Academic Editor

PLOS Genetics

Kirsten Bomblies

Section Editor

PLOS Genetics

Reviewer's Responses to Questions

**Comments to the Authors:**

Reviewer #1: Church et al. describe changes in gene expression across a panel of the Drosophilidae. This was enjoyable to read and a well thought through manuscript. An interesting finding was that species variation overwhelms tissue as you move out in evolutionary time. I think this is an important contrast in the literature to many of the mammalian organ studies.

Major comments:

Some comments in the discussion about why this does look different than many of the mammalian studies is warranted. Generation time, population size, ect. What else do the authors think? Would you expect this finding in all species if you go out far enough in evolutionary time?

Minor comments:

1. The font on many of the figures is really small (Figure 2 is an example).

2. The methods from ref 44 should be summarized in the Methods section (pg 17).

3. Estimated times of divergence would be nice in Figure 1A – gave some perspective on how this compares with other, similar, analyses.

Reviewer #2: Overview:

In this work, the authors present an analysis of tissue-biased gene expression in a phylogenetic context for head and ovary relative to carcass in female Hawaiian Drosophila from twelve species. Four primary analyses are presented, with the results from the correlation analysis made available as a resource via github as in an interactive and queryable format. First, expression is compared across species to identify genes with conserved ovary-biased (or head-biased) expression. Second, an analysis comparing sources of variation, tissue and species, is conducted in order to identify the phylogenetic distance at which species differences within tissue exceed tissue differences within species. Third, gains/losses of tissue bias are examined based on scaled changes in tissue bias (relative to predicted ancestral bias) along the phylogeny. Finally, the Pearson correlation of scaled changes in tissue bias between genes is examined with the goal of identifying co-evolutionary changes in gene expression that may indicate functional relationships between genes. These pairwise correlations are made available to support an approach in which genes showing shared patterns of expression changes may be classified as putatively interacting – a proposed resource for future hypothesis testing.

The main strength of this manuscript is the novel data set and focus on evolution of tissue-biased expression. Collection of this dataset represents a large amount of effort and resources and is of interest to the evo devo community. The manuscript itself is centered around the dataset and presenting an exploration of several ways that evolution of tissue-biased gene expression can be examined in the context of phylogenetic history. Another theme is a more quantitative approach to caveats related to the use of model organism data to make inferences across large evolutionary distances. The focus on tissue bias in a phylogenetic context has the potential to yield novel findings and be of interest to a broad audience.

There is no doubt that these data are a valuable contribution to understanding evolution in the Hawaiian Drosophila and expression evolution at the tissue level more broadly, however there are significant weaknesses in the current approach and manuscript. The largest issue is a lack of the ability to assess confidence in the information content and practical application of the correlation analysis. The small average difference between groups based on known/unknown functional relationships and the large degree of variation are acknowledged by the authors, but there is not an attempt to further assess if/how this can be a useful tool. Some options would be to try alternative analytical approaches, to examine Pearson correlation in a network context more quantitatively, and/or to use validation to assess the usefulness of this approach in identifying candidates. More grounding in the literature via citation and discussion would also aid in highlighting the relevance of the findings and clarify their context in relation to current scientific knowledge. In addition, findings that are relevant to the unique biology and evolutionary history of the Hawaiian Drosophilidae and of the ovary are not a major theme and would be of broad interest.

Major Comments:

1. The pairwise correlation analysis revealed a significant marginal signal of known gene interactions. However, the amount of variation and large overlap between correlations for pairs with unknown relationships versus known is likely to mean that the use for developing hypotheses is minimal. That being said, there are published approaches for analysis of expression co-evolution that could be adapted to tissue bias, alternatives to Pearson’s correlation that might prove to have better properties, and network context that may be relevant for evaluating observed correlations. For example, it is possible that noise and confounding signals (due to complexity of tissue samples) may vary in impact for different pathways, this could be evaluated and might allow for some filtering of informative/non-informative relationships. Partial correlation analysis might be used to explore issues of spurious correlation. There is also a lot of room for the development of an entirely novel approach. Ultimately, experimental validation may be needed to show the relevance of these relationships to identifying candidates with enough efficiency to be a feasible method of identifying good candidates.

2. It is challenging to biologically interpret changes in tissue bias for homology groups. Is it possible to use data from other species groups (where you could create a transcriptome with similar challenges – but have a better handle on ground truth) to better understand what happens in different cases (e.g. a paralog has increased expression in the ovary)? How will the expression estimates behave, more generally, when the number of member transcripts changes along a lineage?

3. I recommend taking a second look at the first paragraph of the introduction and, manuscript-wide, the approach to citation. I know it is not deliberate, but the overall takeaway was an impression of devaluing the existing literature - rather than showing the novel aspects and advantages of your data set and analytical approach as intended. Similarly, in the discussion contextualization of the findings in relation to the existing literature is mostly 'big picture', and that made it difficult to understand the overall contribution being made. For example, how do the findings fit into what has been found in relavent vertabrate studies?

Minor comments.

1. It seems like there might be a need to use assembly and/or mapping approaches designed to be robust with respect to sequence variability present among wild-caught individuals. Was this explored, and were specific approaches used to handle the sequence variability?

2. Ovary-specific vs. ovary-bias – this should be clearly defined if the intent is that specific means something different from bias; otherwise, use bias throughout.

3. I probably missed something, but I wasn’t completely clear on the significance of the specific crossover point. Is it just a way to get a quantitative readout of whether an apples-to-apples comparison exists for tissue bias across a given phylogenetic distance?

4. The behavior of which source of variance, tissue/species, predominates at a particular phylogenetic distance is likely to be affected by different degrees of complexity of the samples. You might expect the species variation to exceed ‘tissue’ at smaller distances when averaging across cell types/tissues in the more complex samples.

5. Pg 16 lines 261-264: With respect to assigning changes to ovary or head versus to the carcass, what was the overall landscape in terms of changes in ovary/head alone and changes in both? Was filtering done based on this?

6. Pg 17 lines 279-281: As mentioned by the authors, ‘head’ is a more complex set of organs, tissues, and cell types. It seems likely that this would reduce the signal for co-evolutionary changes in expression between interacting genes.

7. Pg 19 lines 848-850: Do the homology groups reduce statistical noise or are they used to reduce systematic error that might result from artifacts produced by the assembly?

8. Pg 19 852-853: What is the expectation here and how can we evaluate the strength of the concordance? Is the concordance driven by single orthologs or low complexity homology groups?

9. How issues of multiple testing are handled is not addressed clearly in the methods.

Reviewer #3: In this manuscript Church et al. explore the evolution of ovary biased gene expression in Hawaiian Drosophila using RNA-Seq to estimate gene expression levels, and comparative methods to identify biases in gene expression by tissue and determine how these may have evolved. I find this to be a very nice study, both in the question asked and the methods used, that those interested in gene expression evolution will find of interest. I have only two concerns, which I don't think require additional experiments or data analyses (at least for point 1) but which would be useful to have a discussion about:

1) Clarification on how replicates were used (if this is stated I missed it, so might need to be more explicit if already in the text). Specifically, the methods state that 4 replicates were used, which is great, except for a single species, which is understandable. But it is not clear to me if gene expression estimates across replicates were kept as individual estimates (for results described in 3.1) or were then averaged for results described in 3.2, 3.3, and 3.4. If the results were averaged for results in 3.2, 3.3, and 3.4 it would be helpful (probably essential) to know what effects averaging might have on the results. For example, the average gene expression value can be the same for genes with very different variances in expression so how does variance in gene expression affect the results, such as the probability that gene gains or losses bias? Are genes with large variance in expression across individuals more likely to change bias or the inverse, or is there no relation?

2) Why reconstruct bias, rather than ancestral gene expression levels from which you re-estimate bias? Again, this relates to the variance in gene expression. From what I understand, the bias estimates are point estimates of single value and do not incorporate information on variation in gene expression levels within species (ie., between individuals). But this seems important, so why not reconstruct gene expression levels for all individuals and species, then estimate bias for individuals and get information on them mean and variance in bias estimates?

3) A minor point, but how can gene expression changes in branches 21 and 13 be inferred, doesn't the method use an unrooted tree?

**Have all data underlying the figures and results presented in the manuscript been provided?**

Reviewer #1: Yes

Reviewer #2: Yes

Reviewer #3: Yes

PLOS authors have the option to publish the peer review history of their article (what does this mean?). If published, this will include your full peer review and any attached files.

Reviewer #1: **Yes: **Courtney Babbitt

Reviewer #2: No

Reviewer #3: **Yes: **Vincent J. Lynch

---

## [Decision Letter · Decision Letter 1]

14 Dec 2022

Dear Dr Church,

Thank you very much for submitting your Research Article entitled 'The evolution of ovary-biased gene expression in Hawaiian Drosophila' to PLOS Genetics.

The manuscript was evaluated at the editorial level and by independent peer reviewers. The reviewers appreciated the attention to an important topic but identified some minor concerns.  Although these concerns are about presentation rather than substance, we ask you address them in a revised manuscript.  Mostly, this is a matter of providing more context and better explanations for the filtering tool, and giving the readers some guidance on how to interpret (and how to not over-interpret) the filtered and unfiltered results shown in the figures.

We therefore ask you to modify the manuscript according to the review recommendations. Your revisions should address the specific points made by the second reviewer. No further external review will be necessary - your revisions will be evaluated at the editorial level.

Yours sincerely,

Artyom Kopp

Academic Editor

PLOS Genetics

Kirsten Bomblies

Section Editor

PLOS Genetics

Reviewer's Responses to Questions

**Comments to the Authors:**

Reviewer #1: I am happy with all of the revisions.

Reviewer #2: Overview:

In the revised manuscript the authors have added a method for filtering outliers, added some text to the introduction and discussion and edited the manuscript for clarity. The changes improve the manuscript, but don't fully alleviate my concerns – however since there is not a lot of additional data for this group, it may take time for additional knowledge to develop that would allow more insight into the biological meaning of the correlation results. This is an important investigation that can be a foundation for this future progress.

If it is possible, providing information that helps users make informed choices about how to use and evaluate the results from the data visualization within the tool would increase the contribution made, especially since trainees and others with less experience with these types of analyses are likely to be among the community that use this tool.

Major Comments:

• The added filtering seems reasonable, does it change the result in terms of the relationships shown in Figure 5? Is there a reason for choosing no to report the outcome of this alternative approach in the manuscript itself? I was not able to evaluate if this had a significant impact or not.

Minor Comments:

• Adding an explanation of when/if to implement the filtering option might be helpful for users of the data visualization tool.

• Using the option seems to result in some genes/families not visible (adgf is an example), and I didn’t find cases where it seemed to make a difference – but I also just spot checked. I am not sure if this is the expected behavior.

Reviewer #3: I have no further comments and believe the authors have successfully addressed each comment.

**Have all data underlying the figures and results presented in the manuscript been provided?**

Reviewer #1: Yes

Reviewer #2: Yes

Reviewer #3: Yes

PLOS authors have the option to publish the peer review history of their article (what does this mean?). If published, this will include your full peer review and any attached files.

Reviewer #1: **Yes: **Courtney Babbitt

Reviewer #2: No

Reviewer #3: **Yes: **Vincent J Lynch

---

## [Editor Report · Decision Letter 2]

9 Jan 2023

Dear Dr Church,

We are pleased to inform you that your manuscript entitled "The evolution of ovary-biased gene expression in Hawaiian Drosophila" has been editorially accepted for publication in PLOS Genetics. Congratulations!

Yours sincerely,

Artyom Kopp

Academic Editor

PLOS Genetics

Kirsten Bomblies

Section Editor

PLOS Genetics

Comments from the reviewers (if applicable):

**Data Deposition**

http://datadryad.org/submit?journalID=pgenetics&manu=PGENETICS-D-22-00788R2

**Press Queries**

---

## [Editor Report · Acceptance letter]

17 Jan 2023

PGENETICS-D-22-00788R2 

The evolution of ovary-biased gene expression in Hawaiian *Drosophila*

Dear Dr Church, 

We are pleased to inform you that your manuscript entitled "The evolution of ovary-biased gene expression in Hawaiian *Drosophila*" has been formally accepted for publication in PLOS Genetics! Your manuscript is now with our production department and you will be notified of the publication date in due course.

With kind regards,

Anita Estes

PLOS Genetics

On behalf of:
